# Research on Control of Stewart Platform Integrating Small Attitude Maneuver and Vibration Isolation for High-Precision Payloads on Spacecraft

**Weichao Chi, He Ma, Caihua Wang and Tianyu Zhao ***

College of Sciences, Northeastern University, Shenyang 100819, China; chiweichao@gmail.com (W.C.); berton0417@163.com (H.M.); wangch160137@163.com (C.W.)

\* Correspondence: zhaotianyu@mail.neu.edu.cn; Tel.: +86-15840111007

**Abstract:** The Stewart platform, a classical mechanism proposed as the parallel operation apparatus of robots, is widely used for vibration isolation in various fields. In this paper, a design integrating both small attitude control and vibration isolation for high-precision payloads on board satellites is proposed. Our design is based on a Stewart platform equipped with voice-coil motors (VCM) to provide control force over the mechanism. The coupling terms in the dynamic equations of the legs are removed as the total disturbance by the linear active disturbance rejection control (LADRC). Attitude maneuver and vibration isolation performance is verified by numerical simulations.

**Keywords:** parallel mechanism; active disturbance rejection control; vibration isolation; attitude maneuver





## 1. Introduction

Spacecraft performing specific detection or communication tasks always have strict requirements for the attitude of the payload, which mainly include attitude maneuver and stability. An attitude maneuver is designed to change the attitude of the payload from one to another, while attitude stabilization exists to overcome the internal and external interference moments to keep the attitude of the payload point to a reference orientation.

When the spacecraft is operating in orbit, it will be interfered with by a variety of loads, including the unloading of jet momentum, the rotation of the attitude actuators, the solar radiation pressure, and thermal effects outside the spacecraft. Among them, flywheels, or control moment gyro (CMG), and other high-speed rotation attitude actuators have become the main vibration sources because of the dynamic defects of the rotating parts. In addition, with the increased flexibility of local systems (such as solar panels), a series of problems caused by low-frequency vibration have become more prominent [1–3]. There are many types of structures for the spacecraft multi degree-of-freedom (MDOF) vibration isolation. The most widely adopted structure is the Gough–Stewart parallel mechanism (the Stewart platform) [4]. The Stewart platform was first proposed by Stewart as a six degree-of-freedom flight simulator [5], which was soon applied to the parallel operation mechanism of robots. The "cubic" structure form in which the legs are orthogonal to each other was proposed by Geng et al. to reduce the coupling effect between the legs [6]. Zhou et al. established an accurate mathematical model of 174 geometric parameters based on the joint quaternion and the D–H parameters of each leg and proposed a new kinematics solution method for the general Stewart platform [7]. In the past 20 years, the Stewart platform has gradually been applied to vibration isolation systems [8–10]. At present, many research teams have developed different Stewart platforms to achieve MDOF vibration isolation of satellite payloads. An application of a typical Stewart platform can be seen in the vibration isolation system developed by Hood Technology and the University of Washington [11]. It adopts a cubic configuration of six-axis active vibration isolation with flexible hinges and a large stroke voice coil actuator. Chi et al. proposed a Stewart platform for vibration isolation and used LADRC to create a robust control system [12].

On the other hand, as satellite attitude dynamics and control technology have become increasingly mature, spacecrafts generally use angular momentum exchange devices as the actuators of attitude control systems. To ensure the high-precision and high-stability orientation of a payload on board, great progress has been made by improving control actuators and attitude sensors and developing advanced control technology [13]. Zhang et al. solved the problem of CMG failure or partial failure by redesigning the CMG bearing [14,15]. Yu et al. used $H_\infty$ and composite control methods to achieve large-angle rapid maneuvering control of flexible spacecraft [16]. Ali [17] and Kawajiri [18] also proposed control methods for rapid maneuvering at large angles.

The attitude control and vibration control problems of spacecraft have long been studied as two independent problems. The introduction of flexible vibration isolation platforms, especially in the use of active and passive hybrid vibration isolators, causes the coupling between vibration isolation and attitude control [19]. In the early 1990s, NASA launched a research program called Controls-Structures Interaction (CSI) to realize the joint design and optimization of the two disciplines of control and structure [20]. Narayan et al. analyzed the dynamics of the momentum wheel on the bracket and other subsystems of the satellite, pointing out how the disturbance caused by the rotation of the momentum wheel cannot be ignored when designing the satellite structure, and providing support for the redesign of the bracket with simplified low-level mathematical models [21]. Zhang Yao et al. analyzed the stability requirements of the PID attitude controller and designed the vibration isolator under a three-parameter model [22].

Although a lot of work has been conducted on the research of attitude maneuver and vibration isolation integrated structures, few associate control coefficients with dynamic characteristics. This paper proposes an integrated design of small-angle maneuvering of payload and high-precision vibration isolation. By using a general six degree-of-freedom (DOF) Stewart platform, the resonance problem is solved collaboratively from two aspects: structure improvement and control compensation. This solution does not need to adjust the attitude of the entire spacecraft so the payload attitude can be adjusted very fast with less energy consumption. The LADRC control system is adopted and designed to eliminate the effects of both internal and external disturbance. The dynamic of the platform is analyzed by a simulation and the influence of the control coefficients on system bandwidth is discussed.

## 2. Platform Design and Dynamics of the Legs

To isolate the vibration source, we first need a six DOF motion mechanism. In this paper, a general Stewart platform shown in Figure 1 is applied for the attitude maneuver and vibration isolation of the payload. The six extensible legs of the platform are connected with linear VCMs as the actuators. The attitude of the payload installed on the upper platform can be adjusted by controlling the length of each leg. The VCMs are parallel with diaphragm springs, which connect the upper and lower platform together and act as the elastic element of the passive vibration isolation subsystem at the same time.

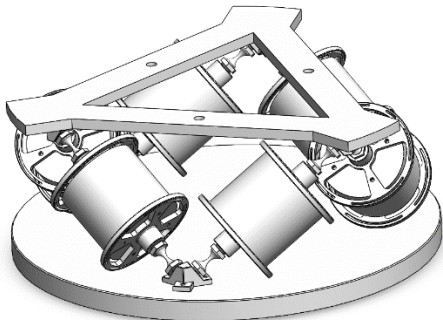

**Figure 1.** The composition of the platform.

As shown in Figure 2, the kinematics and dynamics of the Stewart platform are studied by defining the local coordinate frames fixed on the lower and the upper platforms as frame B and P, respectively. The coordinate frames fixed on the *i*-th upper and lower legs are denoted by $U_i$ and $D_i$, respectively. The position vectors of the ends of the *i*-th leg are obtained as:

$$\mathbf{t}_{pi} = \mathbf{t}_p + \mathbf{p}_i \tag{1}$$

$$\mathbf{t}_{bi} = \mathbf{t}_b + \mathbf{b}_i \tag{2}$$

where $\mathbf{t}_b$ and $\mathbf{t}_p$ are the position vectors of B and P in the inertial frame O. Subtracting $\mathbf{t}_{pi}$ and $\mathbf{t}_{bi}$, the vector of the *i*th leg can be expressed as:

$$\mathbf{l}_i = \mathbf{t}_{pi} - \mathbf{t}_{bi} = \left(\mathbf{t}_p + \mathbf{p}_i\right) - \left(\mathbf{t}_b + \mathbf{b}_i\right) \tag{3}$$

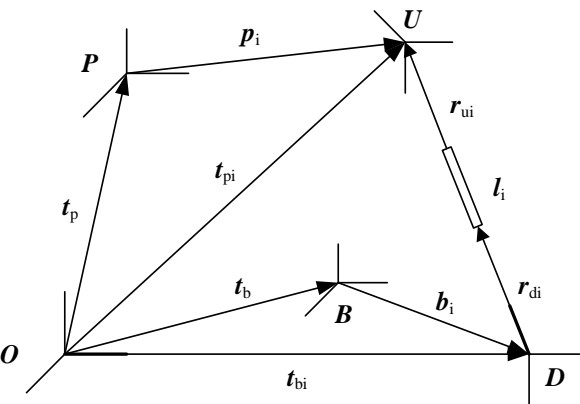

**Figure 2.** Position vectors of the local frames.

Differentiate the above formula, we can view the velocity of the *i*th leg as:

$$
\begin{aligned}
\dot{\mathbf{l}}_i &= \left(\dot{\mathbf{t}}_{pi} - \dot{\mathbf{t}}_{bi}\right) \cdot \boldsymbol{\tau}_i = \left[\left(\dot{\mathbf{t}}_p + \boldsymbol{\omega}_p \times \mathbf{p}_i\right) - \left(\dot{\mathbf{t}}_b + \boldsymbol{\omega}_b \times \mathbf{b}_i\right)\right] \cdot \boldsymbol{\tau}_i \\
&= \begin{bmatrix} \boldsymbol{\tau}_i^T & (\mathbf{p}_i \times \boldsymbol{\tau}_i)^T \end{bmatrix} \begin{bmatrix} \dot{\mathbf{t}}_p \\ \boldsymbol{\omega}_p \end{bmatrix} - \begin{bmatrix} \boldsymbol{\tau}_i^T & (\mathbf{b}_i \times \boldsymbol{\tau}_i)^T \end{bmatrix} \begin{bmatrix} \dot{\mathbf{t}}_b \\ \boldsymbol{\omega}_b \end{bmatrix}
\end{aligned} \tag{4}
$$

where $\boldsymbol{\tau}_i = \mathbf{l}_i / l_i$ is a unit vector, $\boldsymbol{\omega}_p$, $\boldsymbol{\omega}_b$ are the angular velocity vectors of the payload and the base, $\dot{\mathbf{t}}_{pi}$ and $\dot{\mathbf{t}}_{bi}$ are the velocity of the top and bottom of the leg and we have:

$$\begin{cases} \dot{\mathbf{t}}_{pi} = \dot{\mathbf{t}}_p + \boldsymbol{\omega}_p \times \mathbf{p}_i \\ \dot{\mathbf{t}}_{bi} = \dot{\mathbf{t}}_b + \boldsymbol{\omega}_b \times \mathbf{b}_i \end{cases} \tag{5}$$

Then rewrite Equation (4) as:

$$\dot{\mathbf{l}} = \mathbf{H}_p^T \begin{bmatrix} \dot{\mathbf{r}}_p \\ \boldsymbol{\omega}_p \end{bmatrix} - \mathbf{H}_b^T \begin{bmatrix} \dot{\mathbf{r}}_b \\ \boldsymbol{\omega}_b \end{bmatrix} \tag{6}$$

where:

$$\mathbf{H}_p = \begin{bmatrix} \boldsymbol{\tau}_1 & \cdots & \boldsymbol{\tau}_6 \\ \mathbf{p}_1 \times \boldsymbol{\tau}_1 & \cdots & \mathbf{p}_6 \times \boldsymbol{\tau}_6 \end{bmatrix}, \mathbf{H}_b = \begin{bmatrix} \boldsymbol{\tau}_1 & \cdots & \boldsymbol{\tau}_6 \\ \mathbf{b}_1 \times \boldsymbol{\tau}_1 & \cdots & \mathbf{b}_6 \times \boldsymbol{\tau}_6 \end{bmatrix} \tag{7}$$

The force vector that is acting on the upper leg from the lower part is:

$$\mathbf{F} = -\mathbf{K}(\mathbf{l} - \mathbf{l}_0) - \mathbf{C}\dot{\mathbf{l}} + \mathbf{F}_a \tag{8}$$

Here $\mathbf{l} = \begin{bmatrix} l_1 & \cdots & l_6 \end{bmatrix}^{\mathrm{T}}$, $\mathbf{F} = \begin{bmatrix} F_1 & \cdots & F_6 \end{bmatrix}^{\mathrm{T}}$. $\mathbf{F}_a = \begin{bmatrix} F_{a1} & \cdots & F_{a6} \end{bmatrix}^{\mathrm{T}}$ is the force vector of the VCMs. $\mathbf{C} = diag \begin{bmatrix} c_{p1} & \cdots & c_{p6} \end{bmatrix}$ is the damping matrix, $\mathbf{K} = diag \begin{bmatrix} k_1 & \cdots & k_6 \end{bmatrix}$ is the stiffness matrix.

Substitute (6) into (8), we will have:

$$\mathbf{F} = -\mathbf{K}(\mathbf{H}_p^{\mathrm{T}} \begin{bmatrix} \dot{\mathbf{r}}_p \\ \boldsymbol{\omega}_p \end{bmatrix} - \mathbf{H}_b^{\mathrm{T}} \begin{bmatrix} \dot{\mathbf{r}}_b \\ \boldsymbol{\omega}_b \end{bmatrix}) - \mathbf{C}\mathbf{H}_p^{\mathrm{T}} \begin{bmatrix} \dot{\mathbf{r}}_p \\ \boldsymbol{\omega}_p \end{bmatrix} + \mathbf{C}\mathbf{H}_b^{\mathrm{T}} \begin{bmatrix} \dot{\mathbf{r}}_b \\ \boldsymbol{\omega}_b \end{bmatrix} + \mathbf{F}_a \qquad (9)$$

According to the basic formula of dynamics, the upper platform satisfies the following expression:

$$M_p\mathbf{g} + \mathbf{F}_w + \sum_{i=1}^{6} \mathbf{F}_{ui} = M_p\mathbf{a}_p \qquad (10)$$

where, $M_p$ is the mass of the upper platform, $\mathbf{F}_w$ is the disturbance force, $\mathbf{a}_p$ is the mass center acceleration, and:

$$\mathbf{a}_p = \ddot{\mathbf{t}}_p + \boldsymbol{\alpha}_p \times \mathbf{r} + \boldsymbol{\omega}_p \times (\boldsymbol{\omega}_p \times \mathbf{r}) \qquad (11)$$

where $\mathbf{r}$ is the position vector of the mass center. The equilibrium equation of the upper platform and the payload is:

$$-\sum_{i=1}^{6} \mathbf{p}_i \times \mathbf{F}_{ui} + \sum_{i=1}^{6} \mathbf{f}_i + \mathbf{M}_w + M_p\mathbf{r} \times \mathbf{g} = \boldsymbol{\omega}_p \times \mathbf{I}_p^*\boldsymbol{\omega}_p + \mathbf{I}_p^*\boldsymbol{\alpha}_p \qquad (12)$$

According to the Euler equation. Here $\mathbf{M}_w$ is the external moments, $\mathbf{f}_i$ is the force of the VCMs acting to the legs, $\mathbf{I}_p^*$ is a transport for $\mathbf{I}_p$ to the center of mass:

$$\mathbf{I}_p^* = \mathbf{I}_p + M_p \left( \mathbf{r}^T\mathbf{r}E_3 - \mathbf{r}\mathbf{r}^T \right) \qquad (13)$$

From Equations (10) and (12), we have:

$$\mathbf{J}_p \begin{bmatrix} \ddot{\mathbf{t}}_p \\ \boldsymbol{\alpha}_p \end{bmatrix} = \mathbf{J}_b \begin{bmatrix} \ddot{\mathbf{t}}_b \\ \boldsymbol{\alpha}_b \end{bmatrix} + \mathbf{H}_p\mathbf{F} - \boldsymbol{\eta} + \begin{bmatrix} \mathbf{F}_{ext} \\ \mathbf{M}_{ext} \end{bmatrix} \qquad (14)$$

where we define:

$$\mathbf{J}_p = \begin{bmatrix} M_p\mathbf{E}_3 & -M_p\tilde{\mathbf{r}} \\ 0 & \mathbf{I}_p^* \end{bmatrix} + \sum_{i=1}^{6} \begin{bmatrix} \mathbf{Q}_{pi} & -\mathbf{Q}_{pi}\tilde{\mathbf{p}}_i \\ -\tilde{\mathbf{p}}_i\mathbf{Q}_{pi} & \tilde{\mathbf{p}}_i\mathbf{Q}_{pi}\tilde{\mathbf{p}}_i \end{bmatrix} \qquad (15)$$

$$\mathbf{J}_b = \sum_{i=1}^{6} \begin{bmatrix} \mathbf{Q}_{pi} & -\mathbf{Q}_{pi}\tilde{\mathbf{b}}_i \\ -\tilde{\mathbf{p}}_i\mathbf{Q}_{pi} & \tilde{\mathbf{p}}_i\mathbf{Q}_{pi}\tilde{\mathbf{b}}_i \end{bmatrix} \qquad (16)$$

$$\begin{bmatrix} \mathbf{F}_{ext} \\ \mathbf{M}_{ext} \end{bmatrix} = \begin{bmatrix} \mathbf{F}_w + M_p\mathbf{g} \\ -\mathbf{M}_w - M_p\mathbf{r} \times \mathbf{g} \end{bmatrix} \qquad (17)$$

$$\mathbf{Q}_{pi} = m_{ui}\boldsymbol{\tau}_i\boldsymbol{\tau}_i^{\mathrm{T}} + \frac{(\mathbf{E}_3 - \boldsymbol{\tau}_i\boldsymbol{\tau}_i^{\mathrm{T}})}{\lambda_i l_i} \left[ m_{ui}\kappa_i(l_i - l_{ui}) + m_{di}l_{di}^2 \right] - \frac{1}{\lambda_i l_i}\tilde{\boldsymbol{\tau}}_i(\mathbf{I}_{di} + \mathbf{I}_{ui})\tilde{\boldsymbol{\tau}}_i \qquad (18)$$

And:

$$\mathbf{Q}_{bi} = \frac{(\mathbf{E}_3 - \boldsymbol{\tau}_i\boldsymbol{\tau}_i^{\mathrm{T}})}{\lambda_i l_i} \left[ m_{di}l_{di} + m_{ui}\kappa_i l_{ui} \right] - \frac{1}{\lambda_i l_i}\tilde{\boldsymbol{\tau}}_i(\mathbf{I}_{di} + \mathbf{I}_{ui})\tilde{\boldsymbol{\tau}}_i \qquad (19)$$

$$\lambda_i = 2l_{ui} + 2l_{ui} - l_i \qquad (20)$$

where $\eta$ is a higher order infinitesimal in the derivation, the calculation "~" transforms the vector $\mathbf{x} = \begin{bmatrix} x_1 & x_2 & x_3 \end{bmatrix}^{\mathrm{T}}$ to:

$$\tilde{\mathbf{x}} = \begin{bmatrix} 0 & -x_3 & x_2 \\ x_3 & 0 & -x_1 \\ -x_2 & x_1 & 0 \end{bmatrix} \tag{21}$$

At last, we can obtain the task-space equation from (14) as:

$$\mathbf{J}_p\ddot{\mathbf{x}}_p + \mathbf{H}_p\mathbf{C}\mathbf{H}_p^{\mathrm{T}}\dot{\mathbf{x}}_p + \mathbf{H}_p\mathbf{K}\mathbf{H}_p^{\mathrm{T}}\delta\mathbf{x}_p = \mathbf{J}_b\ddot{\mathbf{x}}_b + \mathbf{H}_p\mathbf{K}\mathbf{H}_b^{\mathrm{T}}\delta\mathbf{x}_b + \mathbf{H}_p\mathbf{C}\mathbf{H}_b^{\mathrm{T}}\dot{\mathbf{x}}_b + \mathbf{H}_p\mathbf{F}_a - \eta + \begin{bmatrix} \mathbf{F}_{ext} \\ \mathbf{M}_{ext} \end{bmatrix} \tag{22}$$

where we define:

$$\mathbf{x}_p = \begin{bmatrix} \mathbf{t}_p \\ \boldsymbol{\theta}_p \end{bmatrix}, \mathbf{x}_b = \begin{bmatrix} \mathbf{t}_b \\ \boldsymbol{\theta}_b \end{bmatrix} \tag{23}$$

Equation (22) shows that the upper platform is a second-order system with a nonlinear term. The attitude of the upper platform is determined by the VCMs, and the dynamic of each leg is highly coupled with other legs. The payload is also affected by the external distributions and the vibration of the lower platform.

## 3. Design for the Integrated Attitude-Vibration Control System

In the attitude and vibration control system, the controller receives the state of the object as feedback information, processes and calculates it in real time according to a certain control law, and finally applies the control force or torque to the object through the actuator. In practical processing, assembly errors and measurement noise cannot be ignored, and a lot of simplification and linearization is performed to establish the dynamic equation of the vibration isolation platform. Although these terms are small, they will affect the accuracy of the control system. All undesired disturbances and uncertainties should be eliminated or compensated. Therefore, LADRC control technology that is not based on an accurate mathematical model is adopted [23,24].

### 3.1. Estimation of the States and the Total Disturbance

In the state observation stage, LADRC adds the total disturbance into system states, removes the disturbance from the system by the linear extended state observer (LESO) and the feedback, so that the feedback design needs not a detailed and accurate mathematical model. The main task of the LESO is to establish an extended state observer, by which the nonlinear terms, unmodeled errors, and external disturbances of the system are estimated as the total disturbance [25]. Different from the traditional high-gain observer, the traditional high-gain observer only observes the state of the system without observing the uncertain factors of the system.

If a second-order system is in its general form, it appears as:

$$\ddot{y} + a_1\dot{y} + a_2 y = b_1\ddot{w} + b_2\dot{w} + b_3 w + bu \quad \mathbf{x}_p = \begin{bmatrix} \mathbf{t}_p \\ \boldsymbol{\theta}_p \end{bmatrix}, \mathbf{x}_b = \begin{bmatrix} \mathbf{t}_b \\ \boldsymbol{\theta}_b \end{bmatrix} \tag{24}$$

where $w$ is the disturbance, $u$ and $y$ are the system input and output, respectively. $a1$, $a2$, $b1$, $b2$, $b3$ and $b$ are parameters that can be inaccurate or unknown. Rewrite (24) as:

$$\ddot{y} = -a_1\dot{y} - a_2 y + b_1\ddot{w} + b_2\dot{w} + b_3 w + (b - b_0)u + b_0 u = b_0 u + f \tag{25}$$

where:

$$f = f_1 + f_2 = [-a_1\dot{y} - a_2 y + (b - b_0)u] + [b_1\ddot{w} + b_2\dot{w} + b_3 w] \tag{26}$$

This denotes the total disturbance. $f_1$ is the internal disturbance, including the model uncertainty and changes within the system, such as structural changes, temperature drift, zero drift, and parameter changes. $f_2$ is the external disturbance, such as given disturbance,

and load disturbance. Defining a new state $x_3 = f$, we can produce a pure integrator chain as:

$$\begin{cases} \dot{x}_1 = x_2 \\ \dot{x}_2 = x_3 + b_0 u \\ \dot{x}_3 = h \\ y = x_1 \end{cases} \tag{27}$$

where $b_0$ is an imprecise estimation of $b$, $x_1 = y$, $x_2 = \dot{x}_1$, $h = \dot{f}$. As with Equation (27), the plant is perturbed by total disturbance $f$. Transform the model into state space form as:

$$\begin{cases} \dot{\mathbf{x}} = \mathbf{A}\mathbf{x} + \mathbf{B}u + \mathbf{E}h \\ y = \mathbf{C}\mathbf{x} \end{cases} \tag{28}$$

where:

$$\mathbf{A} = \begin{bmatrix} 0 & 1 & 0 \\ 0 & 0 & 1 \\ 0 & 0 & 0 \end{bmatrix}, \mathbf{B} = \begin{bmatrix} 0 \\ b_0 \\ 0 \end{bmatrix}, \mathbf{C} = \begin{bmatrix} 1 & 0 & 0 \end{bmatrix}, \mathbf{E} = \begin{bmatrix} 0 \\ 0 \\ 1 \end{bmatrix} \tag{29}$$

Next, we use the following observer, which is known as the LESO, to estimate the states that contains the total disturbance:

$$\begin{cases} e = y - z_1 \\ \dot{z}_1 = z_2 + \beta_1 e \\ \dot{z}_2 = z_3 + \beta_2 e + b_0 u \\ \dot{z}_3 = \beta_3 e \end{cases} \tag{30}$$

where $z_1$, $z_2$ and $z_3$ are estimations of $y$, $\dot{y}$ and $f$, respectively.

For Equation (30), the gains $\beta_1$, $\beta_2$ and $\beta_3$ are chosen to ensure the eigenvalues of $(A - LC)$ are in the left complex plane. Gao proposed a method that associates control parameters with bandwidth by assigning all the observer eigenvalues at $-\omega_0$, where $\omega_0$ denotes the bandwidth of the observer. Equivalently, the gain vector is:

$$L = \begin{bmatrix} \beta_1 & \beta_2 & \beta_3 \end{bmatrix} = \begin{bmatrix} 3\omega_o & 3\omega_o^2 & \omega_o^3 \end{bmatrix} \tag{31}$$

This observer also plays the role of a low-pass filter.

### 3.2. Removing the Total Disturbance by Feedback Linearization

In the control stage, we mainly use linear state error feedback control (LSEF), by selecting appropriate control coefficients to eliminate the total disturbance in the feedback process. According to the feedback linearization method, we can offset the total disturbance by simply defining the controller as:

$$u = \frac{-z_3 + u_0}{b_0} \tag{32}$$

where we need to determine the error feedback $u_0$ by substituting (32) into the system of (28):

$$\ddot{y} = (f - z_3) + u_0 = \bar{e}_3 + u_0 \tag{33}$$

where $\bar{e}_3$ is the estimating error of $z_3$. The estimation error can be ignored if the observer is approximately treated as an ideal observer. Then, the relationship between $y$ and $u_0$ becomes a simple linear double-integrator:

$$\ddot{y} \approx u_0 \tag{34}$$

Hence internal and external disturbances are estimated and removed together as the total disturbance. Define $u_0$ as:

$$u_0 = k_p(-z_1 + r) - k_d z_2 \tag{35}$$

where $r$ is the reference value of the tracking. Choose the PD parameters as:

$$k_d = 2\xi\omega_c, k_p = \omega_c^2 \tag{36}$$

where $\xi$ is the damping ratio for reducing oscillation, and $-\omega_c$ is a parameter to be tuned. The form of PD controller in (35) places all the closed-loop poles at $-\omega_c$ without a zero. $\omega_c$ stands for the controller bandwidth. With this controller, the output signal $y$ behaves as the reference value under the controller as:

$$u = -\frac{k_p}{b_0}z_1 - \frac{k_d}{b_0}z_2 - \frac{1}{b_0}z_3 + \frac{k_p}{b_0}r \quad k_d = 2\xi\omega_c, k_p = \omega_c^2 \tag{37}$$

In summary, the diagram of LADRC is shown as Figure 3.

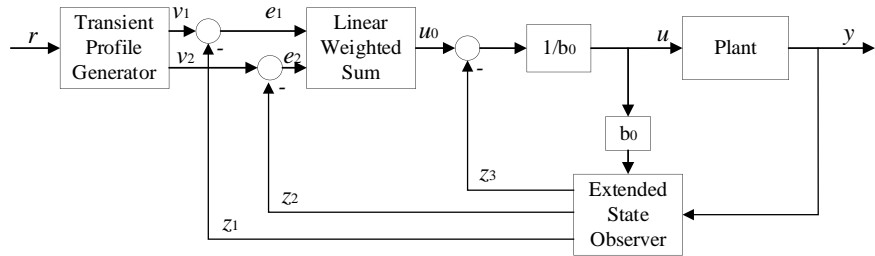

**Figure 3.** The diagram of LADRC system.

### 3.3. Decoupling Control of the Legs

As discussed in Section 2, the dynamics of the six legs of the Stewart platform are strongly coupled. This problem is also solved by LADRC under the idea of the total disturbance. Consider System (22) in the multi-input-multi-output (MIMO) form:

$$\begin{cases} \ddot{\mathbf{x}} = \mathbf{f}(x, \dot{x}, w, \dot{w}, t) + \mathbf{B}\mathbf{u} \\ \mathbf{y} = \mathbf{x} \end{cases} \tag{38}$$

where:

$$\mathbf{x} = \begin{bmatrix} x_1 & x_2 & \cdots & x_6 \end{bmatrix}^T \tag{39}$$

$$\mathbf{f} = \begin{bmatrix} f_1 & f_2 & \cdots & f_6 \end{bmatrix}^T \tag{40}$$

$$\mathbf{u} = \begin{bmatrix} u_1 & u_2 & \cdots & u_6 \end{bmatrix}^T \tag{41}$$

$$\mathbf{B} = \begin{bmatrix} b_{11} & \cdots & b_{16} \\ \vdots & \vdots & \vdots \\ b_{61} & \cdots & b_{66} \end{bmatrix} = \mathbf{J}_p^{-1}\mathbf{H}_p \tag{42}$$

If $\mathbf{B}$ is reversible, let $\mathbf{U} = \mathbf{B}\mathbf{u}$, for the $i$-th channel we have:

$$\begin{cases} \ddot{x}_i = f_i(x, \dot{x}, \cdots, x_6, \dot{x}_6, t) + U_i \\ y_i = x_i \end{cases} \tag{43}$$

where $\mathbf{U} = \begin{bmatrix} U_1 & U_2 & \cdots & U_6 \end{bmatrix}$ is the virtual control matrix. The element $U_i$ and the output $y_i$ of each channel are totally decoupled. The coupling between different legs and

the external disturbances are treated as the total disturbance and are removed together by the feedback. The actual control vector **u** can be determined by:

$$\mathbf{u} = \mathbf{B}^{-1}\mathbf{U} \tag{44}$$

The diagram of a MIMO system controlled by LADRC is shown in Figure 4. Here the coupling term can be removed by LADRC as the disturbance of matrix B. The precise parameters of B are not necessary if the control parameter $\omega_0$ and $b_0$ are well tuned.

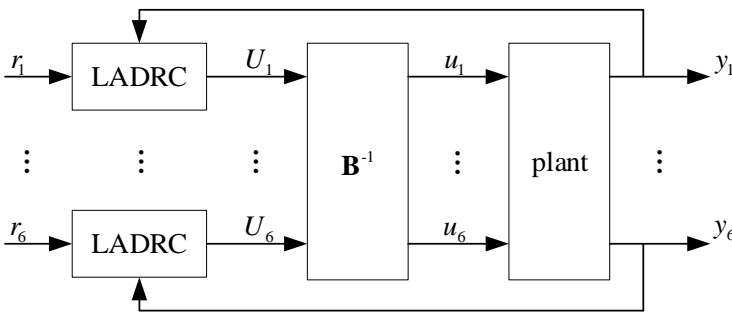

**Figure 4.** The diagram of a MIMO system controlled by LADRC.

## 4. Results and Discussion

The model of the integrated 6-DOF attitude maneuver and vibration isolation system is built by SIMULINK to estimate the performance of the LADRC algorithm. The main specifications for the model are shown in Table 1. This section presents two simulations: payload attitude maneuver and vibration isolation realized by the Stewart platform.

**Table 1.** Main specifications of the platform model.

| Specification | Value |
| --- | --- |
| Upper platform radius | 0.165 m |
| Height | 0.079 m |
| Legs length | 0.162 m |
| Stiffness of diaphragm spring | $4.8 \times 10^4$ N/m |
| Mass of top platform | 3.29 kg |
| Mass of payload | 4.85 kg |

### 4.1. Attitude Maneuver Performance

The simulation studied a payload attitude maneuver along three axes. A reference angle $\theta = 0.01$ rad is given at 0.1 s. The initial value of $\mathbf{x}_p$ is set to $\begin{bmatrix} 0 & 0 & 0 & 0 & 0 & 0 \end{bmatrix}^T$. A high-frequency noise in the attitude measurements of 1000 rad/s is artificially imposed as sensor output. The LADRC parameters for the controller are chosen as:

$$\begin{cases} b_0 = 1 \\ \omega_{oi} = 200 \quad , i = 1, \cdots, 6 \\ \omega_{ci} = 100 \end{cases} \tag{45}$$

The time-domain response of the three attitude angles is shown is Figure 5a. The payload complete attitude adjustment is 0.3 s, and there is no vibration in the steady state. It also presented the impact of sensor noise in Figure 5b. For comparison with Figure 5a, the observer bandwidth $\omega_o$ is tuned to 1000 as the hypothetical noise frequency, which causes a residual oscillation with a magnitude of $0.055 \times 10^{-3}$ rad. It can be noticed that $\omega_o$ can decide the control speed and impact of the noise to the system performance. If $\omega_o$ is set in the range of the noise band, the steady state error tends to be higher and the system output oscillates although the attitude adjusts faster. If $\omega_o$ is set to be lower than the noise

frequency, the noise is filtered and the system output is more smooth. Figure 5c shows another approach of speeding up the maneuver process by increasing the value of the controller bandwidth $\omega_c$ to 200. It is shown that, although the adjusting period is shortened, the maximum control force totals 3.5 times that of under the parameters (a). It requires a higher performance of the actuators and is sometimes hard to realize in engineering.

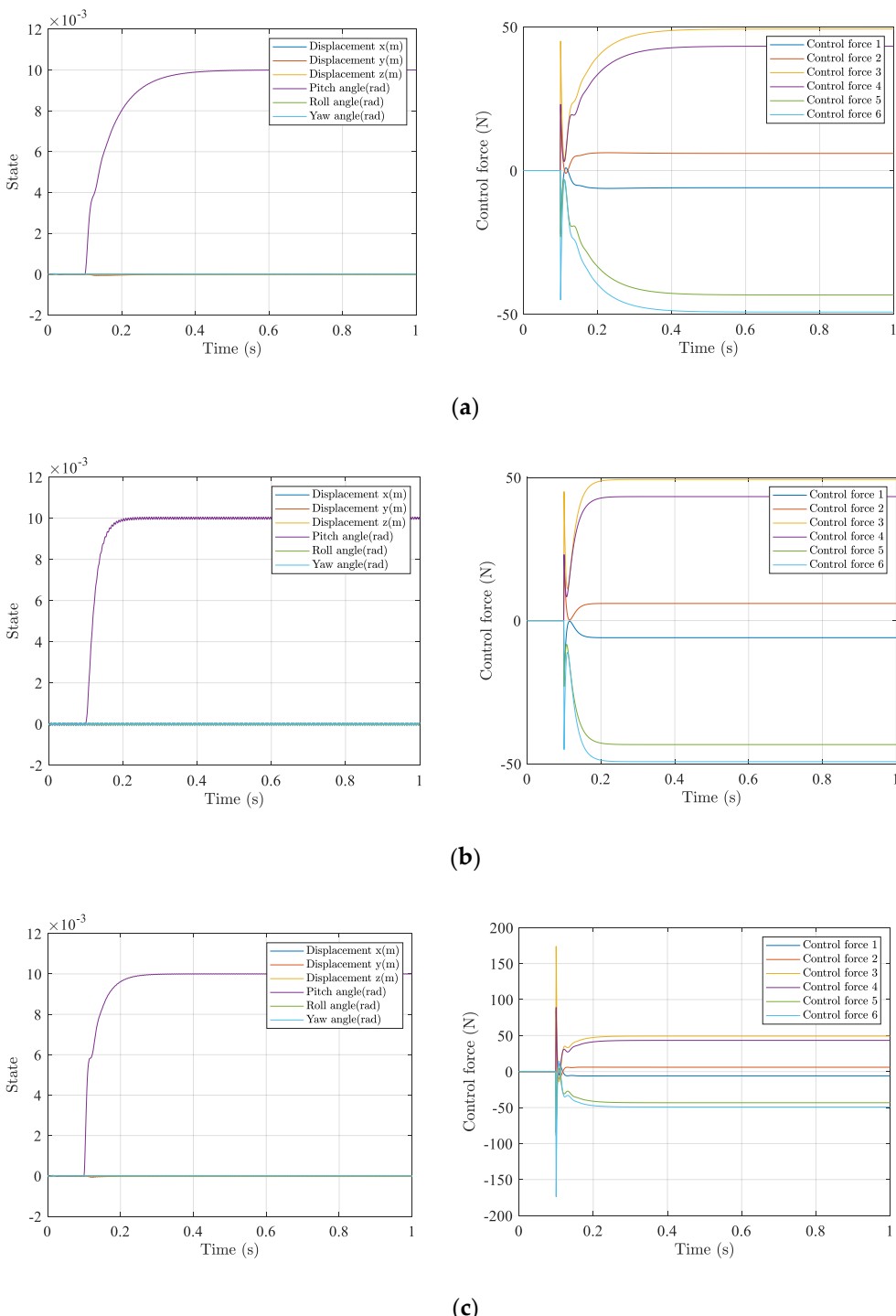

**Figure 5.** The time history of the pitch angle and control force with different $\omega_o$. (**a**) $\omega_o = 200, \omega_c = 100$. (**b**) $\omega_o = 1000, \omega_c = 100$. (**c**) $\omega_o = 200, \omega_c = 200$.

### 4.2. Vibration Isolation Performance

If the reference attitude vector coincides with the inertial frame O, the issue is reduced to vibration isolation. The effectiveness of the LADRC method is evaluated through the numerical simulation in Simulink. The same parameters are used as in Section 4.1. The sinusoidal displacement disturbance along the x-axis is given at the base platform. The amplitude of the disturbance is $10^{-4}$ m and the frequency is the resonance frequency of the system at 60.6 rad/s.

Figure 6 show that the LADRC controller reduces the vibration response along the x-axis from $1.7 \times 10^{-3}$ to $1.0 \times 10^{-5}$. The response along other axes performs similarly with a different attenuating range. The vibration isolation performance can be affected by the LADRC parameters because it determines the adjustment speed of the control system. Higher observing and controlling bandwidth can result in a better vibration isolation, but more sensitive to noise. Thus, the control parameters must be set in a compromise between attitude adjusting speed and sensitivity to the noise, considering the frequency of vibration and noise.

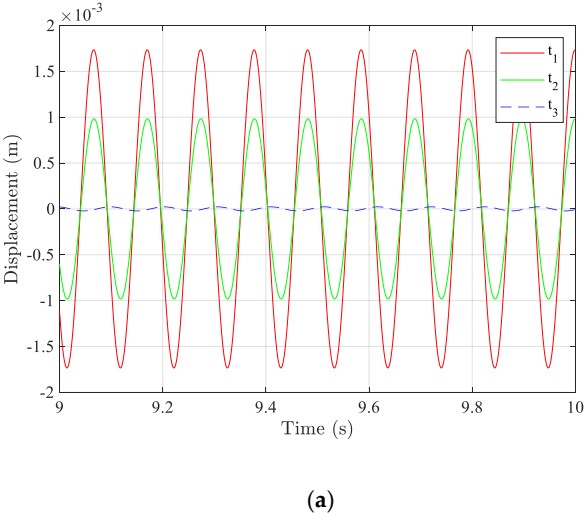

**(a)**

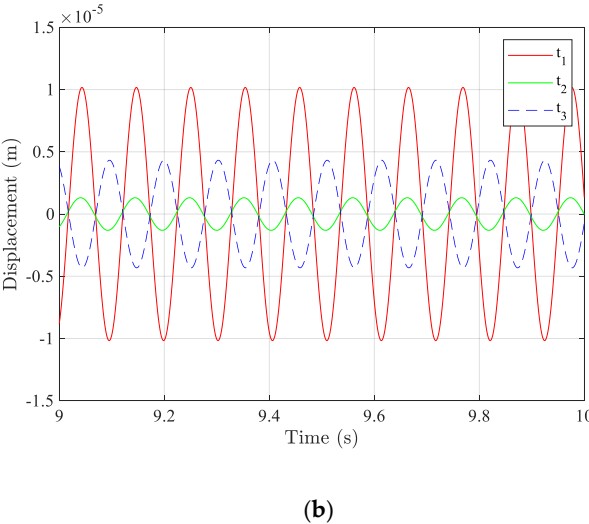

**(b)**

**Figure 6.** The response of sinusoidal disturbance. (**a**) Open-loop. (**b**) Under LADRC controller.

## 5. Conclusions

An integrated design of small-angle maneuvering and vibration isolation of payload is proposed in this paper by using a general six DOF Stewart platform. The dynamic

equations of the Stewart platform are discussed, and the control system based on the LADRC algorithm is designed. The advantage of this algorithm is to remove the total disturbance by simple linear feedback, by observing the unmodeled dynamics, nonlinear terms and the coupling dynamics as the internal disturbance, even if the control structure is not accurately modeled. The proposed control system is shown to perform efficiently in the numerical simulations and the relationship between control parameters and system bandwidth and performance is discussed.

**Author Contributions:** Conceptualization, W.C. and T.Z.; methodology, W.C.; software, W.C.; validation, W.C., H.M., C.W. and T.Z; investigation, W.C.; writing—original draft preparation, W.C.; writing—review and editing, H.M., C.W. and T.Z. All authors have read and agreed to the published version of the manuscript.

**Funding:** This research was funded by the National Natural Science Foundation of China, grant number 12002080.

**Acknowledgments:** We would like to thank Jianqiao Sun (University of California, Merced) for his advice on this research.

**Conflicts of Interest:** The authors declare no conflict of interest.

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
