# Peer review of "Research on Control of Stewart Platform Integrating Small Attitude Maneuver and Vibration Isolation for High-Precision Payloads on Spacecraft"

_aerospace, doi:10.3390/aerospace8110333_

Round 1
Reviewer 1 Report
Dear Authors,
I will recommend this paper for its publication if you can address the next points:
- The Reviewer is not sure about the novelty of this work. There are many works which have studied similar problems. I suggest to add/extend a paragraph in the introduction that shows the merits of this work in the context of what other researchers have done.
- Figure 1 looks come from: Chi, W.; Cao, D.; Wang, D.; Tang, J.; Nie, Y.; Huang, W. Design and Experimental Study of a VCM-Based Stewart Parallel Mechanism Used for Active Vibration Isolation. Energies 2015, 8, 8001-8019. https://doi.org/10.3390/en8088001. Please, improve the quality of the figure.
- Line 151: review the sentence: “All undesired disturbances and should [..]“
- Section 4.2 shows a problem which has been widely studied in the literature: Vibration Isolation by using a Stewart Platform. It is important to compare the effectiveness of this method against other controller. At least, mention it. This part of the work does not show any novelty.
- In Section 4.2, Authors describe the VI performance. The results show x-axis performance. Do the authors consider to study the VI performance in other axes?
- The Reviewer suggests to show some bode graphs to analyse the VI performance of the proposed controller.
- Have the Authors considered the interaction of the isolator (SP) with the supporting structure? Recent work show important effects when this interaction is considered.
- Some important references are missing, those which come from authors such as: McInroy and A. Preumont.
- Conclusions Section should be improved. The novelty of the work compared with other works should be highlighted.
Author Response
Dear Editors,
On behalf of my co-authors, we thank you very much for giving us an opportunity to revise our manuscript. We appreciate editors and reviewers very much for their positive and constructive comments and suggestions on our manuscript entitled “Research on Control of Stewart Platform Integrating Attitude Maneuver and Vibration Isolation for Spacecraft”.
We have studied reviewer’s comments carefully and have tried our best to revise our manuscript according to the comments. The revision is marked in red in the manuscript. The detail point-to-point reply is listed at the bottom of this letter.
Once again, we would like to express our great appreciation to you and the reviewer for comments on our paper. Looking forward to hearing from you.
Thank you and best regards.
Yours sincerely,
Editor and Reviewer comments:
Reviewer #1:
I will recommend this paper for its publication if you can address the next points:
The Reviewer is not sure about the novelty of this work. There are many works which have studied similar problems. I suggest to add/extend a paragraph in the introduction that shows the merits of this work in the context of what other researchers have done.
Reply:
We have added a brief conclusion on what is needed to improve in the work of other researchers in line 67-70 and clarified the innovation about the association between LADRC coefficient and bandwidth in line 78-79.
Figure 1 looks come from: Chi, W.; Cao, D.; Wang, D.; Tang, J.; Nie, Y.; Huang, W. Design and Experimental Study of a VCM-Based Stewart Parallel Mechanism Used for Active Vibration Isolation. Energies 2015, 8, 8001-8019. https://doi.org/10.3390/en8088001. Please, improve the quality of the figure.
Reply:
We have replaced the figure with a new assembly drawing with more details in line 87.
Line 151: review the sentence: “All undesired disturbances and should [..]“
Reply:
The sentence has been revised as “All undesired disturbances and uncertainties should be eliminated or compensated” in line 158.
Section 4.2 shows a problem which has been widely studied in the literature: Vibration Isolation by using a Stewart Platform. It is important to compare the effectiveness of this method against other controller. At least, mention it. This part of the work does not show any novelty.
Reply:
Thank you for your advice. We are very prudent for comparing the effectiveness between different control methods because in the theoretical and simulation research, the performance of the system can be affected by inappropriate parameter selection. So, what we focus is the characteristic that the LADRC has clear physical meaning, be independent of the mathematical model and easy to realize in engineering.
In Section 4.2, Authors describe the VI performance. The results show x-axis performance. Do the authors consider to study the VI performance in other axes?
Reply:
Yes, the structure has the ability to isolate vibration in 6 degree of freedom, the response of payload in x-axis to the disturbance from the base in the same axis is obvious and representative. Performance in other axes is similar as that in x-axis. Besides, we are going to have experimental study for this condition in future. So, we take one direction to illustrate the performance.
The Reviewer suggests to show some bode graphs to analyse the VI performance of the proposed controller.
Reply:
The bode graph is studied in our previous work about the frequency characteristics of the vibration isolation system. As this paper is not focus on the same issue, so we did not show the bode diagram this time.
Have the Authors considered the interaction of the isolator (SP) with the supporting structure? Recent work show important effects when this interaction is considered.
Reply:
The interaction with the supporting structure is not considered in this paper. The main purpose of this paper is to present the application of LADRC on the Stewart platform for attitude and vibration control. We are considering to have a further research about this issue in future.
Some important references are missing, those which come from authors such as: McInroy and A. Preumont.
Reply:
The research of McInroy and Preumont are remarkable, we have added references from line 335 to 338.
Conclusions Section should be improved. The novelty of the work compared with other works should be highlighted.
Reply:
Conclusions has been revised to highlight the novelty of our work in line 302-307.
Reviewer #2:
This paper focuses on an interesting application of an integrated attitude and vibration control strategy on a Stewart platform.
However, English language should be widely revised, beginning from the paper abstract, where repetitions ("mechanism", etc) and syntax ("an integrating design of attitude maneuver and vibration isolation for the payload on spacecraft is proposed" could be better written as maybe "a design integrating both attitude control and vibration isolation for payloads on board satellites", etc). Other sentences should be improved, such as lines 21-23, line 24 "satellite working" in "satellite operating" and so on. Please revise carefully the overall English of the paper.
Reply:
Thank you for your advice! We have make changes according to your suggestions at line 9-11 line 25 and other places which are marked in red.
Concerning the references in the introduction, please add more up-to-date papers related to Stewart platform active control methods. Also, more attention should be given in the introduction to answer the question "how does this paper contribute to the field with respect to the current state-of-the-art?"
Reply:
We have added some references and a brief conclusion on what is needed to improve in the work of other researchers in line 335 to 338, and clarified the innovation about the association between LADRC coefficient and bandwidth in line 78-79.
Line 121, please clarify what Mw, fi are. Line 122, I* is defined as "a translation", and then you talk about "centroid", please clarify that I* is a moment of inertia (the "translation", or better the "transport", is achieved by using Huygens theorem) and define what is "centroid", as it was not mentioned before (it is the center of mass, but please define it). Line 129, what is η7? What is λi? Please clarify carefully all the terms in the equations, or add a reference to a previous paper in which the basic formulation was developed and explained, if any.
Reply:
The definition of Mw, fi and λi has been added in line 127. is a higher order infinitesimal in the derivation, which is also clarified after the formulas in line 140.
Revise line 149. Line 159 add a reference for the observer. Line 218 "coupled".
Reply:
The sentences have been revised, a reference for the observer is added in line 167.
Fig. 5 (b): please provide the order of magnitude of the residual oscillations in the state.
Reply:
The magnitude of the residual oscillations is provided in line 268.
As general suggestion, I would encourage the authors to better specify in the abstract and introduction that the strategy they are proposing is limited to very small attitude maneuver and also precision vibration issues, so to stress that the system is thought to be applied in high-precision applications (such as optics, etc), and maybe to specify it also in the title.
Reply:
The points of small attitude maneuver and high-precision applications are specified in the title, abstract and introduction (line 2-4, 10-12, 71-72).
Reviewer 2 Report
This paper focuses on an interesting application of an integrated attitude and vibration control strategy on a Stewart platform.
However, English language should be widely revised, beginning from the paper abstract, where repetitions ("mechanism", etc) and syntax ("an integrating design of attitude maneuver and vibration isolation for the payload on spacecraft is proposed" could be better written as maybe "a design integrating both attitude control and vibration isolation for payloads on board satellites", etc). Other sentences should be improved, such as lines 21-23, line 24 "satellite working" in "satellite operating" and so on. Please revise carefully the overall English of the paper.
Concerning the references in the introduction, please add more up-to-date papers related to Stewart platform active control methods. Also, more attention should be given in the introduction to answer the question "how does this paper contribute to the field with respect to the current state-of-the-art?"
Line 121, please clarify what Mw, fi are. Line 122, I* is defined as "a translation", and then you talk about "centroid", please clarify that I* is a moment of inertia (the "translation", or better the "transport", is achieved by using Huygens theorem) and define what is "centroid", as it was not mentioned before (it is the center of mass, but please define it). Line 129, what is η7? What is λi? Please clarify carefully all the terms in the equations, or add a reference to a previous paper in which the basic formulation was developed and explained, if any.
Revise line 149. Line 159 add a reference for the observer. Line 218 "coupled".
Fig. 5 (b): please provide the order of magnitude of the residual oscillations in the state.
As general suggestion, I would encourage the authors to better specify in the abstract and introduction that the strategy they are proposing is limited to very small attitude manoeuvre and also precision vibration issues, so to stress that the system is thought to be applied in high-precision applications (such as optics, etc), and maybe to specify it also in the title.
Author Response

(The authors gave the same response as above.)

Round 2
Reviewer 1 Report
Dear Authors,
I've recommended this paper for its publication.
Kind regards,
The Reviewer
Reviewer 2 Report
The comments have been addressed. Still some minor English corrections are needed.